# Associations of Overall Survival with Geriatric Nutritional Risk Index in Patients with Advanced Pancreatic Cancer

**DOI:** 10.3390/nu14183800

**Published:** 2022-09-15

**Authors:** Christina Grinstead, Thomas George, Bo Han, Saunjoo L. Yoon

**Affiliations:** 1Department of Biobehavioral Nursing Science, College of Nursing, University of Florida, 1225 Center Drive, Gainesville, FL 32610-0187, USA; 2Department of Medicine, Division of Hematology and Oncology, College of Medicine, University of Florida, Gainesville, FL 32611, USA; 3Department of Surgery, Keck School of Medicine, University of Southern California, Los Angeles, CA 90007, USA

**Keywords:** pancreatic cancer, nutritional status, survival, body weight, albumin, geriatric nutritional risk index

## Abstract

Nutritional assessment is critical in cancer care to maintain quality of life and improve survival. The Geriatric Nutritional Risk Index (GNRI) may be a practical tool to assess nutritional status and predict survival. This study aimed to examine survival using GNRI in advanced-stage pancreatic cancer (PC). The retrospective analysis used data of patients with stage III or IV PC. Inclusion criteria: age > 18 and hospital admission for at least three days at or following diagnosis between 2014 and 2017. Data collected: demographics, albumin levels, BMI and weight. Days between the first and last admission, median survival and GNRI scores calculated. Patients categorized into groups: any nutritional risk (GNRI ≤ 98) and no nutritional risk (GNRI > 98). 102 patients had a median survival of 87.5 days and mean GNRI of 98.7. Patients surviving longer than 90 days showed higher mean weight (*p* = 0.0128), albumin (*p* = 0.0002) and BMI (*p* = 0.0717) at the first admission. Mean survival days for patients at any nutritional risk were 110 days compared to 310 days for no nutritional risk (*p* = 0.0002). GNRI score at first admission after diagnosis is associated with survival. It is vital to monitor nutritional status using weight and albumin to promote increased survival from diagnosis.

## 1. Introduction

Pancreatic cancer (PC) is one of the most deadly cancers, with a 5-year relative survival rate of only 11% after diagnosis [1]. Efforts to improve survival among patients with PC have had limited success, in part due to the lack of valid and reliable assessment tools. Many patients do not often receive the care needed to extend life as healthcare practitioners face struggles in identifying those most at risk of developing adverse outcomes, making it challenging to implement treatments and interventions early enough to be effective [2]. 

Nutritional risk assessment is critical in patients with pancreatic cancer as metabolic dysfunction, unintentional weight loss and systemic inflammation are commonly found to result from tumor-induced and treatment-associated changes in physiological function [3,4]. The development of cachexia, a debilitating condition, was reported in more than 63% of patients with pancreatic cancer [3,4]. Cachexia is defined as progressive, involuntary weight loss (a weight loss of >5% over 6 months or >2% if a BMI < 20 kg/m^2^), loss of skeletal muscle mass and systemic inflammation [5], which results in decreased survival, increased toxicity of treatment, increased length of hospitalization and poor quality of life [5]. Tumor host factors affecting appetite, adipose tissue, muscle mass and treatment-induced symptoms contribute to the malnutrition and unintentional weight loss commonly found in cachexia and patients with pancreatic cancer [6]. Patients with pancreatic adenocarcinoma presenting cachexia during disease progression have more appetite loss, fatigue, diarrhea and nausea than their non-cachectic counterparts [3]. While cachexia is characterized by weight loss and muscle wasting due to hypermetabolism, systemic inflammation and imbalance between protein synthesis and degradation, not solely due to decreased calorie intake [5], these patients also present symptoms associated with maldigestion and malabsorption of nutrients [7] in addition to muscle wasting. This suggests the importance of monitoring nutritional status to decrease treatment toxicities, lower the incidence of hospitalization, enhance the quality of life and improve overall survival [8].

Currently, cachexia is often assessed by measuring skeletal muscle mass using computed tomography (CT) and magnetic resonance imaging (MRI) [9] as well as other serological biomarkers such as C-reactive protein, lymphocytes and transferrin [10]. Although concepts of cachexia and nutritional impairment are not equal and cachexia cannot be reversed by nutritional support [5], proper nutritional support is considered fundamental to manage cachexia [11]. Accurate assessment of nutritional status is essential to intervene early and lower or delay the risk of adverse outcomes; however, there is currently no standardized method to assess nutritional status efficiently and effectively in patients with pancreatic cancer. These measures have been shown to be associated with worse prognosis and survival as well as a variety of other negative outcomes in multiple studies and are often easily accessible through the collection of routine clinical data among pancreatic cancer patients [9,10].

The Geriatric Nutritional Risk Index (GNRI) may also be useful in determining the nutritional status of patients with pancreatic cancer and predicting survival utilizing clinically available data, as it is calculated using serum albumin and weight [12]. The GNRI [12] originated from the Nutritional Risk Index (NRI) and was developed to be practical for nutritional assessment in older adults [13]. This scale has been validated extensively in several patient populations, including dialysis, heart failure and cancer patients and shown to be associated with overall survival predictability [14,15,16,17]. A predictive value of GNRI has been tested in patients younger than 65 years old, although it has traditionally been used in adults over 65 [18,19]. The GNRI, however, has not been fully validated for patients with pancreatic cancer.

The purpose of this study is to examine associations between nutritional status and overall survival in patients with advanced pancreatic cancer using commonly available clinical data as indicators of cancer progression. We hypothesize that (1) mean serum albumin, weight, body mass index (BMI) and GNRI scores at the time of diagnosis will be lower in the short-term survival group compared to the longer-term survival group, (2) weight, albumin levels, mean BMI and BMI categories will be positively associated with survival and (3) patients at nutritional risk based on the GNRI score will have shorter overall survival compared to those at no nutritional risk among patients with advanced pancreatic cancer.

## 2. Materials and Methods

### 2.1. Specific Aims

Three specific aims are as follows: (1) examine albumin, weight, BMI and GNRI scores at the time of diagnosis in patients with stages III and IV pancreatic cancer and determine the median survival days to categorize the short-term and long-term survival, (2) compare the differences in weight, BMI and albumin levels between the short-term survival and long-term survival groups, (3) compare survival days between pancreatic cancer patients with any nutritional risk and no risk based on GNRI scores [9] and examine the predictive value of GNRI in patients with late-stage of pancreatic cancer (stages III and IV).

### 2.2. Study Population

This retrospective analysis was conducted using the de-identified data extracted from the electronic health records of patients with pancreatic cancer hospitalized between 2014 and 2017 at a single large institution in the southeast of the United States. Inclusion criteria are patients 18 years or older diagnosed with stages III or IV pancreatic cancer hospitalized for at least 3 days at or following diagnosis. The study is IRB approved (IRB#201703076). 

### 2.3. Demographic and Clinical Variables

Demographic variables include age, race and sex. race were categorized into White and Non-White. Non-White includes African Americans, Asians and other ethnicities due to the small number of patients in this sample. Clinical variables include tumor location (head, body and tail), cancer stage (III or IV), BMI (kg/m^2^), weight (Kg) and albumin levels (g/dL). We extracted the clinical data of BMI, weight and albumin levels at the first hospital admission after diagnosis from the electronic health records. BMI values were further categorized into groupings defined as underweight (BMI < 20 kg/m^2^), normal weight (20 kg/m^2^ ≤ BMI < 25 kg/m^2^), overweight (25 ≤ BMI <30 kg/m^2^) and obese (BMI ≥ 30 kg/m^2^). 

### 2.4. Survival Days

We calculated survival days by measuring the number of days between the first hospital admission after diagnosis and the last admission before death or at the time of death, whichever comes first as the primary endpoint. Because the exact dates of death were not available for all patients in the electronic health records, the last recorded hospitalization was considered the primary endpoint before death. We categorized patients into two groups for further inferential analysis based on median survival days. Short-term survival was defined as 90 days or less and long-term survival as longer than ninety days of survival. 

### 2.5. GNRI

GNRI scores were calculated based on serum albumin, current weight and *ideal weight (WLo)* using the following formula: GNRI = [1.489 × albumin (g/L)] + [41.7 × (weight/WLo)] [12]. The Lorentz formula was used to determine the *ideal weight (WLo)* as follows: for men: H − 100 − [(H − 150)/4] and for women: H − 100 − [(H − 150)/2.5]. When weight exceeded WLo, the ratio was set to 1. Nutritional risk categories are *major* risk (GNRI < 82), *moderate* risk (82 ≤ GNRI < 92), *low* risk (92 ≤ GNRI ≤ 98) and *no* risk (GNRI > 98) [12]. After categorizing all patients into four GNRI groups, they were re-categorized into two groups (i.e., *any* risk (GNRI ≤ 98) and *no* risk (GNRI > 98)) for inferential analysis. 

### 2.6. Statistical Analysis

Descriptive statistics were used to describe the demographic and clinical variables using the mean, median, standard deviation, range, frequency and percentage when it is applicable. Simple linear regression analysis was conducted to calculate the association between survival days as a continuous outcome variable and age as a continuous predictor variable. T-tests were used to compare mean survival days between two groups. ANOVA was used to compare race, tumor location, BMI category and GNRI group with Bonferroni adjustment to decrease false-positive results when multiple comparisons. Chi-squared tests were used to determine associations of group membership with survival days. 

After grouping patients based on median survival and GNRI scores, logistic regression, chi-square tests and ANOVA with Bonferroni adjustments were used. Logistic regression was used to examine age with group membership as a binary outcome measure. Chi-square tests determined associations between categorical variables. ANOVA was used to test the significance of mean differences between more than two groups. Significance was set at *p* < 0.05.

## 3. Results

### 3.1. Demographics and Clinical Characteristics

For the total sample, median survival was 87.5 days (mean 216 days; range 1–1726 days). The mean age was 65.6 years; ranges 38–85), 58% were male and 85% were White (Table 1). Stage III and stage IV cancers were 28.4% and 71.6%, respectively. At the time of admission, the cohort had a mean weight of 82 kg (40.8–129.4), mean serum albumin of 3.9 g/dL (1.7–5.0), mean BMI of 28.1 kg/m^2^ (16.3–54.7) and a mean GNRI score of 98.7 (66.5–116.2). The GNRI scores indicated that 2.0% of the patients were at major risk, 15.3% at moderate risk, 25.5% at low risk and 57% at no risk. BMI indicated that 8.8% were underweight, 27.5% normal weight, 31.4% overweight and 32.4% were obese at the time of the first admission after diagnosis (Table 2).

### 3.2. Associations with Survival

Univariate analysis demonstrated that neither age (*p* = 0.0608), sex (*p* = 0.4496), race (*p* = 0.3912), or tumor location (*p* = 0.6078) was significantly associated with median overall survival (OS). Stage (*p* = 0.0351) did have an inverse relationship with OS. Clinical characteristics indicated a positive association between weight (*p* = 0.0101), albumin (*p* <0.0001), GNRI score (*p* < 0.0001) and overall survival at the time of the first admission after diagnosis. Mean BMI (*p* = 0.0840) and BMI categories (*p* = 0.8267) were not associated with overall survival.

### 3.3. Comparing Survival Groups

Patients were categorized into one of two groups based on the median survival of 87.5 days. We defined that Group 1 survived for 90 days or less (n_1_ = 51) and group 2 survived for over 90 days (n_2_ = 51) between the first and last admission. Percentages of stage III and stage IV cancer were 15.7% and 84.3% for group 1, while group 2 consisted of 41.2% and 58.8%, respectively. Group 1 was 84.3% White and 51% male with a mean age of 68, while group 2 was 86.3% White and 67.7% male with a mean age of 64. For group 1, findings indicated a mean weight of 76.8 ± 21.2 kg, albumin level of 3.7 g/dL, BMI of 26.8 kg/m^2^ and GNRI of 95.6 at the first admission. Group 2 indicated a mean weight of 87.3 ± 21.0 kg, albumin level of 4.1 g/dL, a BMI of 29.4 kg/m^2^ and a GNRI of 101.5 at the first admission. BMI in Group 1 consisted of 25.5% obese, 36.2% overweight, 27.7% normal weight and 10.6% underweight patients, while group 2 consisted of 37.3% obese, 33.3% overweight, 21.6% normal weight and 7.8% underweight patients. In group 1 (survival days ≤ 90 days), 57.5% had *any risk* of GNRI scores (GNRI ≤ 98) and 42.5% with *no risk* (GNRI > 98). In group 2, 29.4% were under *any risk* of GNRI scores, while 70.6% demonstrated *no risk*. The difference in GNRI scores between group 1 and group 2 was significant (*p* = 0.0045). Age (*p* = 0.0378) and stage (*p* = 0.0078) were statistically significant differences between the two groups, while no differences were found in sex (*p* = 0.22), race (*p* = 0.07) and tumor location (*p* = 0.50). Clinical characteristics at the time of the first admission after diagnosis showed higher weight (*p* = 0.0128), albumin (*p* = 0.0002) and GNRI (*p* = 0.0003) to be associated with survival greater than 90 days. There was no difference in mean BMI between the two groups, with group 1 having a mean BMI of 26.8 kg/m^2^ and BMI of 29.4 kg/m^2^ in group 2 (*p* = 0.0717), nor the categorized BMIs (e.g., normal, high BMI) between the two groups (*p* = 0.71). However, there were statistically significant differences (*p* = 0.0045) in nutritional risks between group 1 and group 2 based on the GNRI scores, with group 1 having 27 patients and group 2 having 15 patients at nutritional risk (Table 2).

### 3.4. Comparisons between Two GNRI Groups

GNRI scores were calculated and analyzed based on the data from 98 of 102 patients, as four patients did not have sex or height or both information necessary for GNRI calculation. Those four patients not included in the GNRI analysis were all in the short-term survival group. Differences in mean and median survival days of the total sample with and without those four patients are shown in Table 1 and Table 2. Initial four GNRI groups were re-categorized into two groups, any risk (GNRI ≤ 98) and no risk (GNRI > 98), with 42.9% of 98 patients being at nutritional risk. The group with any nutritional risk had a mean age of 66, was 54.8% male and 85.7% White while the no nutritional risk group had a mean age of 65 and was 62.5% male and 83.9% White. Stages III and IV patients were 28.6% and 71.4% in both any risk and no risk groups, respectively. Median survival for any risk group (GNRI ≤ 98) was 38.5 days and 176.0 days for no risk group (GNRI > 98), while the mean survival days were 109.8 days and 309.6 days, respectively. The survival days were significantly different (*p* = 0.0002) between the two GNRI groups with shorter time survival in any risk group. No statistically significant differences were found in age (*p* = 0.5875), sex (*p* = 0.5342), race (*p* = 0.8075), tumor location (*p* = 0.12) or stage (*p* = 1) (Table 3).

## 4. Discussion

Our findings indicated that patients with advanced pancreatic cancer demonstrating any nutritional risk (GNRI scores ≤ 98) had significantly decreased median and mean survival days regardless of demographics, stage and tumor location. Conversely, patients with no nutritional risk (GNRI > 98) had three times longer mean survival days and 4.5 times longer median survival days than patients with any nutritional risk. In addition, albumin level and body weight showed positive associations with survival. These findings underpin the importance of nutritional assessment and potential tailored intervention in supportive care for patients with pancreatic cancer to improve survival days even in their advanced stages. 

Coupled with statistical significance, the findings of our study imply clinical significance, considering the high prevalence of cachexia among patients with pancreatic cancer [6,20,21]. As unintentional weight loss, systemic inflammation and poor nutritional status are hallmarks of cachexia and are prevalent in this disease, monitoring weight, nutritional risk and albumin are critical in the care of patients with pancreatic cancer to halt cachexia development and progression [22,23]. Monitoring albumin levels from the time of diagnosis and beyond has clinical utility. Albumin is positively associated with nutritional status and BMI independent of systemic inflammation and liver function. Hypoalbuminemia is associated with a higher risk of nutritional impairment, lower overall survival and higher inflammatory markers (e.g., CRP) [24,25] since inflammation and malnutrition decrease albumin concentration by reducing albumin synthesis [26]. Multiple studies have elucidated its importance throughout pancreatic cancer progression [22,23,27]. Weight as a modifiable factor has been shown to affect patient outcomes significantly and should be aggressively managed during treatment [21,28,29].

As the GNRI is a valuable tool in assessing nutritional status, our findings are consistent with previous studies that demonstrated the association of the GNRI with prognosis in colorectal cancer, pancreatic cancer and lymphoma [15,16,18]. Our study corroborates the importance of monitoring weight, albumin and GNRI from diagnosis. Further research, however, warrants determining the cause-and-effect relationship between these factors and survival to confirm their clinical relevance in improving survival and developing interventions to improve albumin synthesis in this cohort more aggressively. The GNRI measured with clinically available weight and albumin levels may predict overall survival and prognosis among patients with pancreatic cancer. Using GNRI is also practical and easily implemented in clinical practice as it can be incorporated into electronic health records (EHR) and automated for calculation with minimal time and effort. 

Nutritional status should be monitored closely for early intervention. GNRI seems to be a practical tool to aid in identifying patients that may be more likely to experience decreased survival times after diagnosis, particularly in patients with PC, who are at high risk of developing cachexia. Nutritional interventions are critical in this patient population, showing that nutritional improvement may counteract muscle degradation and weight loss in cachexia [7,8,30,31,32]. Multi-modal interventions are recommended when appropriate for patients if it is applicable [32]. Various nutritional supplementation will provide the necessary nutrients required to build and stabilize muscle and tissue and increase energy stores that can help restore the energy imbalance caused by hypermetabolism [32]. However, it is vital to start nutritional intervention early at the time of diagnosis before patients develop cachexia, considering that already developed cachexia is challenging to reverse. Pancreatic enzyme insufficiencies caused by tumor growth or surgical resection can prevent oral intake of nutrition, but recent studies have shown that pancreatic enzyme replacement therapy may be successful in improving the absorption of nutrients and maintaining weight for those suffering from this complication [33]. Enteral or parenteral nutrition can supplement patients with poor intake or a possible alternative that can allow the delivery of nutrients, although temporary [30]. 

Patients in our study had advanced pancreatic cancer (stages III and IV) with a median survival of 87.5 days and a median age of 66 years old. Our study population had slightly older and shorter median survival days compared to a recent study conducted by Xu et al. [34], who included patients with stages from I to IV pancreatic cancer, which might result in a longer median survival time of 20.9 months and a slightly younger median age of 61 years compared to our study sample [34]. Previous studies have shown that increased age is associated with decreased survival time [2,35,36], while our study does not show survival differences based on age. It is well known that having more advanced stages of cancer has been shown to correlate with decreased survival. Our study finding is consistent with the existing knowledge about survival and the late stage of pancreatic cancer.

There were limitations to this study. First, other factors contributing to survival in pancreatic cancer were not available for analysis, including tumor size, liver metastasis, liver functional reserve, treatment modalities, lifestyle and overall health [2]. We recognize that, due to this limitation, the use of the GNRI alone is not sufficient in the assessment of nutrition among pancreatic cancer patients. Additionally, clinically relevant biomarkers related to nutritional status, prognosis and survival, such as measuring muscle mass and quality and serological measures, were not available for analysis. This will lead us to conduct future studies with more variables to validate our findings with current methods of assessment. However, our study provides important information showing that the GNRI is associated with survival regardless of other mitigating factors and previously validated biomarkers, suggesting the empirical evidence of nutritional assessment using the clinically available variables. Third, our survival days were defined as the number of days between the first admission after diagnosis and the last admission prior to death or documented death date on EHR, which needed caution when interpreting the findings. Due to the survival days showing the length of time to hospitalization resulting in death rather than the date of death specifically, the generalization of findings may be limited to patients with PC requiring a more intensive level of treatment that can no longer be cared for effectively in an outpatient setting. Fourth, the existing data analysis limited us to considering potential confounding factors that might provide more information. The generalizability of the findings is also a limitation because data were from one tertiary care institution in the southeast United States with a relatively modest sample size. Finally, the combination of both stages III and IV patients in the analysis may be a limitation due to the heterogeneity of the sample, knowing that the median survival of patients with stage III may be longer than that of stage IV.

However, our study has multiple strengths. First, the study findings corroborated the significance of monitoring nutritional status in the care of patients with pancreatic cancer, who are highly vulnerable to cachexia and shortened survival. Second, the study provided evidence of the utility of GNRI with clinically available variables. Third, implementation of GNRI is calculation is feasible and scalable, particularly with hospital EMR data which should be able to transition to the outpatient setting easily. Utilization of GNRI at hospital admission is also supported by the opportunity for nutritional counseling services, diagnosis of pancreatic insufficiency and dietary education/recommendations as a component of the hospital stay. Lastly, the study indicated the magnitude of early detection of nutritional risk and interventions. Future research is warranted to examine the additional factors and confounders that may affect nutritional risk and to develop personalized interventions using the GNRI as a detection or monitoring tool with the goal of increasing survival and maximizing quality of life. 

In summary, nutritional risk, as measured using weight and albumin is significantly associated with survival among patients with pancreatic cancer. Patients at no nutritional risk survive almost three times longer than those at nutritional risk. More research is needed to incorporate such tools into clinical practice and improve survival for this patient population.

## Figures and Tables

**Table 1 nutrients-14-03800-t001:** Demographics and survival days between two survival groups.

Variables	Total Sample (N = 102)	Group 1 ^†^ (n_1_ = 51)	Group 2 ^‡^ (n_2_ = 51)	*p*-Values (*p*< 0.05)
**Survival (Days)**				
Mean (±SD)	215.6 ± 303.9	30.4 ± 24.9	400.7 ± 340.6	
Median	87.5	26.0	288.0	
Range	(1–1726)	(1–84)	(91–1726)	
**Age**				
Mean (+SD)	66 + 9.8	68 + 8.5	64 + 10.7	**0.0378**
Range	(38–85)	(48–85)	(38–84)	
**Sex ***				
Male	58 (58.0%)	25 (51.0%)	33 (64.7%)	0.22
Female	42 (42.0%)	24 (49.0%)	18 (35.3%)	
**Race ^#^** White African American Asian Other				
	87 (85.3%)	43(84.3%)	44 (86.3%)	0.07
	12 (11.8%)	5 (9.8%)	7 (13.7%)	
	2 (2.0%)	2 (3.9%)	0 (0%)	
	1 (1.0%)	1 (2.0%)	0 (0%)	
	33 (32.4%)	14 (27.5%)	19 (37.3%)	

* 2 missing data points; ^†^ Group 1: Survival < 90 days; ^‡^ Group 2: Survival > 90 days, ^#^
*p*-value based on white vs. all others

**Table 2 nutrients-14-03800-t002:** Clinical characteristics between two survival groups.

Variables	Total Sample (N = 102)	Group 1 ^†^ (n_1_ = 51)	Group 2 ^‡^ (n_2_ = 51)	*p*-Values (*p* < 0.05)
**Tumor Location**				
Head	52 (52%)	24 (49.0%)	28 (54.9%)	0.50
Body	16 (16%)	6 (12.2%)	10 (19.6%)	
Tail	14 (14%)	8 (16.3%)	6 (11.8%)	
Overlapping	18 (18%)	11 (22.5%)	7 (13.7%)	
**Stage**				
III	29 (28.4%)	8 (15.7%)	21 (41.2%)	**0.0078**
IV	73 (71.6%)	43 (84.3%)	30 (58.8%)	
**At First Admission**				
Weight (Kg)				
Mean (+SD)	82.0 ± 21.7	76.8 ± 21.2	87.3 ± 21.0	**0.0128**
Range	(40.8–129.4)	(40.8–129.4)	(48.4–127.1)	
Albumin (g/dL)				
Mean (+SD)	3.9 ± 0.55	3.7 ± 0.53	4.1 ± 0.52	**0.0002**
Range	(1.7–5.0)	(1.7–4.6)	(2.9–5.0)	
BMI (Kg/m^2^)				
Mean (+SD)	28.1 ± 7.2	26.8 ± 6.4	29.4 ± 7.8	0.0717
Range	(16.3–54.7)	(17.2–48.2)	(16.3–54.7)	
GNRI				
Mean (+SD)	98.7 ± 8.4	95.6 ± 8.2 *	101.5 ± 7.6	**0.0003**
Range	(66.5–116.2)	(66.5–110.2)	(84.9–116.2)	
GNRI Categories ***				
Any Risk	42 (42.9%)	27 (57.5%)	15 (29.4%)	**0.0045**
No Risk	56 (57.0%)	20 (42.5%)	36 (70.6%)	
BMI Categories				
Underweight	9 (8.8%)	5 (9.8%)	4 (7.8%)	0.71
Normal Weight	28 (27.5%)	16 (31.4%)	12 (23.6%)	
Overweight	32 (31.4%)	16 (31.4%)	16 (31.4%)	
Obese	33 (32.4%)	14 (27.5%)	19 (37.3%)	

*** 4 missing data points due to lack of information on sex and/or height; ^†^ Group 1: Survival ≤ 90 days; ^‡^ Group 2: Survival > 90 days.

**Table 3 nutrients-14-03800-t003:** Clinical Characteristics and GNRI categories.

Variables	Total Sample (N = 98)	Any Risk * (n_1_ = 42)	No Risk ** (n_2_ = 56)		*p*-Values (*p* < 0.05)
**Survival (Days)**					
Median	100.0	**38.5**	**176.0**		**0.0002**
Mean (±SD)	224.0 + 307.2	**109.8 + 139.3**	**309.6 + 366.8**		
Range	(2–1726)	(3–540)	(2–1726)		
**Age**					
Mean (+SD)	65.7 + 9.8	66 + 9.5	65.0 + 10.0	0.5875	
Range	(38–84)	(45–81)	(38–84)		
**Sex**					
Male	58 (59.2%)	23 (54.8%)	35 (62.5%)	0.5342	
Female	40 (40.8%)	19 (45.2%)	21 (37.5%)		
**Race *****					
White	83 (84.7%)	36 (85.7%)	47 (83.9%)	0.8075	
African American	12 (12.2%)	5 (11.9%)	7 (12.5%)		
Asian	2 (2.0%)	1 (2.4%)	1 (1.8%)		
Other	1 (1.0%)	0 (0%)	1 (1.8%)		
**Tumor Loc.**					
Head	51 (52.0%)	23 (54.8%)	28 (50%)	0.12	
Body	16 (16.3%)	3 (7.1%)	13 (23.2%)		
Tail	14 (14.3%)	6 (14.3%)	8 (14.3%)		
Overlapping	17 (17.4%)	10 (23.8%)	7 (12.5%)		
**Stage**					
III	28 (28.6%)	12 (28.6%)	16 (28.6%)	1.0	
IV	70 (71.4%)	30 (71.4%)	40 (71.4%)		

GNRI: Geriatric Nutritional Risk Index; * Any risk = GNRI ≤ 98; ** No risk = GNRI > 98; *** *p*-value based on white vs. all others.

## Data Availability

The data presented in this study are available on reasonable request from the corresponding author.

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
