# Peer review of "Associations of Overall Survival with Geriatric Nutritional Risk Index in Patients with Advanced Pancreatic Cancer"

_nutrients, 2022, doi:10.3390/nu14183800_

Round 1
Reviewer 1 Report
Despite the fact that the authors conducted a correct observational retrospective study and well described and selected the model and statistical methods, unfortunately the selection of the parameters they took for the study does not bring anything new and revealing. If the authors have access to patient data (electronic health records, IRB approved), they may be tempted to find parameters defining e.g. muscle metabolism or other non-obvious and unexplored compounds or factors.
Reviewer 2 Report
Thanks for the opportunity to review. I have the following minor edits:
Page 1 Line 33-34 delete available and clinical practice
Page 1 line 44 change to increased length of hospitalization
Page 2 Line 49 change to cachectic
Page 2 line line 50 delete in this population
Page 2 line 63 add predictability after survival
Page 7 line 252 add Enteral or parenteral nutrition
Reviewer 3 Report
In the present paper the Authors intended to investigate the association of the GNRI with the overall survival of patients affected with stage III and IV pancreatic cancer. The assessment of nutritional status is very important in all patients with cancer, especially in those with cancer of the gastrointestinal tract that are at high risk to develop cachexia and that have the poorest prognosis in terms of survival. The early detection of malnutrition in the pre-cachectic phase could prevent or delay the progression of this syndrome to more advanced stages. In this regard, the first step in the early detection of malnutrition is the implementation and validation of screening tools to identify patients at nutritional risk. With this purpose, a variety of tools already exists, but GNRI has not been validated yet in patients with pancreatic cancer, thus rendering the present work original.
In general, the paper is well written, the aims are clear and properly addressed and the conclusions are consistent with the results obtained, including remarks about study limitations.
However, some major points are missing in this study:
1) not only BMI, but also body weight loss and skeletal muscle index are very important to assess the nutritional risk. It is very well known and documented that low muscle mass is associated with reduced survival in patients with pancreatic cancer and other cancer types. Importantly, low muscle mass is part of the diagnostic criteria to define pre cachectic and cachectic patients in association with body weight loss and body mass index. In addition to low muscle mass, low muscle quality characterized by fatty infiltration, termed as myosteatosis, is a predictor of poor outcomes after resection of various malignancies including pancreatic cancer. In addition CT image analyses reveal low levels of muscle also in individuals who are overweight or obese (sarcopenic obesity), thus helping in a more appropriate stratification of the cohort of patients that seems to be mostly of normal weight or overweight. Even if the information of body weight loss may not be available from the electronic records, CT images are routinely acquired in the standard care of cancer patients and can provide information on body composition. The authors should perform the analyses of CT scans and include information about skeletal muscle mass and fat to better characterize the cohort of subjects and to investigate their association with GNRI.
2) In addition to that, other serological markers have been validated and are currently used to assess malnutrition, also in oncologic patients: transthyretin, retinol binding protein, transferrin, creatinine/height index and lymphocytes’count. To strengthen the role of GNRI to assess nutritional risk, it would be important to test for the association of GNRI also with these biomarkers, not only with BMI and albumin. The authors should check the electronic records for those clinical variables and make additional statistical analyses. Usually, at least lymphocytes, transferrin and creatinine are included in the routinary panel of tested biomarkers.
Minor point:
Table 1 is too long, it would be better to split it in two Tables: Table 1 with anthropometric and tumor data, Table 2 with clinical variables and GNRI or BMI categories.
Round 2
Reviewer 3 Report
The fact that additional parameters (skeletal muscle index and other biomarkers of malnutritions) are not available in the clinical records and therefore cannot be included in further analyses, is a limitation of the present study. Therefore, this point must be discussed in the Discussion section. In addition to that, I strongly suggest to include in the Introduction some statements (and the relative references) regarding the importance of muscle mass and quality as prognostic factor for survival of pancreatic cancer patients as well as other validated serological biomarkers in the assessment of malnutrition, in order to give a more complete background for the contextualization the study.
